# Valine–Niclosamide for Treatment of Androgen Receptor Splice Variant-Positive Hepatocellular Carcinoma

**DOI:** 10.3390/cancers17152535

**Published:** 2025-07-31

**Authors:** Emma J. Hoelzen, Hanna S. Radomska, Samuel K. Kulp, Adeoluwa A. Adeluola, Lauren A. Granchie, Jeffrey Cheng, Anees M. Dauki, Moray J. Campbell, Shabber Mohammed, Enming Xing, Min Hai, Mayu Fukuda, Xiaolin Cheng, Mitch A. Phelps, Pui-Kai Li, Christopher C. Coss

**Affiliations:** 1Division of Pharmaceutics and Pharmacology, College of Pharmacy, The Ohio State University, Columbus, OH 43210, USA; hoelzen.1@osu.edu (E.J.H.); radomska.1@osu.edu (H.S.R.); kulp.1@osu.edu (S.K.K.); adeluola.1@osu.edu (A.A.A.); granchie.1@osu.edu (L.A.G.); hai.29@osu.edu (M.H.); fukuda.17@osu.edu (M.F.); phelps.32@osu.edu (M.A.P.); 2Division of Medicinal Chemistry and Pharmacognosy, College of Pharmacy, The Ohio State University, Columbus, OH 43210, USA; cheng.1545@osu.edu (J.C.); mohammed.210@osu.edu (S.M.); xing.244@osu.edu (E.X.); cheng.1302@osu.edu (X.C.); li.27@osu.edu (P.-K.L.); 3Exelixis Inc., Alameda, CA 94502, USA; dauki.1@osu.edu; 4Samuel Oschin Comprehensive Cancer Institute, Cedars-Sinai Medical Center, Los Angeles, CA 90048, USA; moray.campbell@cshs.org

**Keywords:** liver cancer, nuclear hormone receptor, androgen receptor splice variant, niclosamide, drug repurposing, hepatocellular carcinoma

## Abstract

Hepatocellular carcinoma (HCC) is the predominant form of liver cancer and a leading cause of global cancer-related mortality. At present, the current front-line systemic therapies for advanced hepatocellular carcinoma offer limited improvements to overall survival. In this study, we repurposed niclosamide, an anti-tapeworm therapeutic, for use in HCC by utilizing an amino acid conjugate, valine–niclosamide. This study provides further evidence of the role of the androgen receptor (AR) and its truncated splice variants (SVs) in HCC and offers valine–niclosamide as a candidate therapeutic that addresses not only AR signaling in HCC but also other known pro-cancer signaling pathways. We believe that these data provide a rationale for the further study of valine-conjugated niclosamide as a broad anti-cancer therapeutic that addresses, in part, niclosamide’s poor oral bioavailability and describe AR-SV-positive advanced HCC as a potential use case for valine–niclosamide.

## 1. Introduction

### 1.1. Hepatocellular Carcinoma

Hepatocellular carcinoma (HCC) is currently the second-leading cause of cancer-related mortality globally. Its incidence has been increasing globally, and, in the United States, it is currently the fastest-growing cause of cancer-related mortality. Despite recent advances in front-line therapeutic options for advanced HCC, which comprises the majority of HCC cases, the five-year survival rate for HCC in the US is less than 12% [1,2]. HCC exhibits sexual dimorphism, with men having an increased rate of both incidence and mortality [3]. Additionally, HCC precursors including steatosis, hepatitis, cirrhosis, and non-alcoholic fatty liver disease (NAFLD), now called metabolic dysfunction-associated steatotic liver disease (MASLD), have a higher incidence in men [4].

### 1.2. Role of Androgen Receptor and Its Splice Variants in Hepatocellular Carcinoma

The androgen receptor (NR3C4/AR) has been implicated in HCC progression and provides a potential explanation for the sexual dimorphism exhibited in both its incidence and mortality. The AR, like other nuclear hormone receptors, functions as a transcription factor. It is bound by androgens, leading to receptor activation, nuclear localization, and dimerization. It then binds to a variety of cofactors and androgen response elements (AREs) within the cellular genome to regulate a diverse set of target genes [4]. While androgens play a role in AR activation, in our recent review, we discuss evidence that the AR, rather than androgens, is implicated in HCC progression and provide alternative mechanisms for AR activation, including mTOR crosstalk, lipogenesis-driven AR activity, CCRK-mediated activity, FAK-mediated signaling, and, most notably, AR splice variant (AR-SV) expression in HCC [3]. AR-SV expression has been described by Dauki et al. in 78% of HCC patients within The Cancer Genome Atlas (TCGA) cohort [5]. These results, in conjunction with the clinical failure of anti-androgen therapy, enzalutamide, in HCC, emphasize the need for HCC therapeutics directly addressing AR protein levels and AR activity [6].

### 1.3. Current Landscape of HCC Therapeutics

The current front-line therapeutic options for advanced HCC have long included sorafenib, a multikinase inhibitor, and have more recently been expanded to include lenvatinib, another multikinase inhibitor, and the combination of atezolizumab/bevacizumab, comprising an immune checkpoint inhibitor (ICI) and anti-VEGF-A antibody. The atezolizumab/bevacizumab combination offers a median overall survival (mOS) improvement of 19.2 months, and current efforts are being directed toward determining appropriate second-line approaches for patients who fail ICI therapies [2]. Most recently, the combination of durvalumab, an anti-PD-L1 ICI, and tremelimumab, an anti-CTLA-4 ICI, successfully completed a phase III trial in unresectable HCC and was approved for front-line use, with an mOS increase of 16.4 months [7]. Considering the significant mortality of the disease and the modest overall survival improvements offered by current front-line systemic therapies, the exploration of additional therapeutic options for use alone and in combination with existing therapies is warranted. Due to the role of AR activity in HCC, there have been several calls for AR therapeutics in HCC that address both AR/AR-SV protein levels and AR activity [3,8,9]. In this study, we propose repurposing the anthelmintic drug niclosamide for use in liver cancer due to its desirable anti-HCC properties and ability to lower AR/AR-SV protein levels.

### 1.4. Niclosamide as an Anti-Cancer Therapeutic

Niclosamide is an anthelmintic drug, approved by the FDA in 1982 for the treatment of tapeworm infections, and is included in the World Health Organization’s (WHO) essential medicines list [10]. Niclosamide’s activity against tapeworms is largely credited to its ability to uncouple oxidative phosphorylation. However, more recent studies investigating niclosamide’s mechanism of action have demonstrated its involvement in targeting a variety of oncogenic pathways [11,12,13]. For this reason, niclosamide has been investigated for a variety of alternative indications, including Parkinson’s disease, diabetes, bacterial infections, viral infections, and cancer [10]. For our particular use case, AR-SV(±) HCC, niclosamide was highlighted in two drug screens for its ability to both reverse HCC gene expression patterns and lower AR-V7 protein, a dominant prostate cancer (PCa) AR-SV [14,15]. These effects, in addition to niclosamide’s activity against NF-κB, STAT3, KRAS, Myc, and mTOR, among many other oncogenic pathways, and its ability to both upregulate p53 and enhance PD-1/PD-L1 blockade, position it well as an oncological multitool [16,17,18,19,20,21,22,23,24,25,26,27]. While niclosamide’s anti-cancer activity provides a strong rationale for cancer indications, its pitfalls as an oral therapeutic serve as the primary hurdle to utilizing niclosamide as a systemic therapy in the clinic. Niclosamide is poorly soluble and poorly orally bioavailable. In a prostate cancer clinical trial, niclosamide was tested due to its activity against AR-SVs but failed to show therapeutic efficacy due to its inability to reach therapeutic plasma levels without dose-limiting gastrointestinal toxicities [28]. Various attempts to improve niclosamide’s bioavailability have been made, including the development of a more soluble niclosamide ethanolamine salt, a self-microemulsion of niclosamide, niclosamide-loaded nanoparticles, and various analogs of niclosamide [14,16,29,30,31,32,33]. Here, we demonstrate that niclosamide analogs maintain anti-AR activity against HCC cell lines expressing AR-SVs and that the amino acid conjugation of niclosamide is an effective means of increasing its solubility and absorption, improving its pharmacokinetic profile for a variety of uses.

## 2. Materials and Methods

### 2.1. Cell Culture and Reagents

Human hepatocellular carcinoma cell lines SNU-423, SNU-475, and HepG2, along with the immortalized human liver epithelial cell line THLE2, were purchased from the American Type Culture Collection (ATCC, Manassas, VA, USA). Primary male hepatocytes were obtained from Lonza (Basel, Switzerland). HCCLM3 cells were received from Dr. Thomas Schmittgen, University of Florida (Gainesville, FL, USA). All cells were regularly tested for *Mycoplasma* using PlasmoTest (rep-pt1, InvivoGen, San Diego, CA, USA) and confirmed negative. Cells were sub-cultured in media based upon the ATCC recommendation: RPMI1640 (SH30027.LS, Cytiva, Marlborough, MA, USA), EMEM (30-2003, ATCC), or BEGM (CC-3170, Lonza, Basel, CHE, Switzerland). Cell media were supplemented with 10% FBS (S1620 (HI), Biowest, Nuaillé, France). Cells were incubated at 37 °C in a humidified atmosphere with 5% CO_2_. Primary male hepatocytes were grown, plated, and cultured as directed by the Lonza “Suspension and Plateable Cryopreserved Hepatocyte” protocol using the Hepatocyte Culture Medium BulletKit (CC-3198, Lonza). Paclitaxel (PTX) and valacyclovir were obtained from MCE (HY-B0015 and HY-17425A, MedChemExpress, Monmouth Junction, NJ, USA). Niclosamide analogs and amino acid-conjugated compounds were synthesized by Dr. Pui-Kai Li’s laboratory, as detailed in Appendix A.

### 2.2. TCGA LIHC AR-SV Transcript Presence and Abundance

AR-SV transcript abundance was calculated utilizing patient data from The Cancer Genome Atlas (TCGA) Liver Hepatocellular Carcinoma (LIHC) cohort [34] and analyzed for AR-SV abundance as previously described [5]. AR-SV transcript abundance was expressed as a percentage of the AR-FL transcript abundance on a per tumor basis, and tumors were stratified into four quartiles, with Quartile 1 (Q1) representing the lowest AR-SV transcript abundance and Quartile 4 (Q4) representing the highest.

### 2.3. siRNA Knockdown

siRNA knockdown of AR was performed utilizing Lipofectamine 2000 (11668019, Invitrogen, Carlsbad, CA, USA), as per the manufacturer’s instructions. Small interfering RNAs targeting AR (M-003400-02 and L-003400-00, Dharmacon, Lafayette, CO and CD.Ri.209684.13.5, Integrated DNA Technologies (IDT), Coralville, IA, USA) were purchased and pooled for use. A non-specific control siRNA duplex (4457287, Ambion, Carlsbad, CA, USA) was utilized as a negative control. siRNAs were transfected into target cells as previously described [5]. Cell protein lysates were collected at 48 h after transfection and subsequently analyzed via Western blotting.

### 2.4. RNA Sequencing and GSEA Analysis

RNA was isolated from siRNA knockdown cells and submitted to the Nationwide Children’s Steve and Cindy Rasmussen Institute for Genomic Medicine (Columbus, OH, USA) for total RNA sequencing. RNA purity and concentration were measured using a NanoDrop Spectrophotometer (ND2000, Thermo Scientific, Waltham, MA, USA). The ScriptSeq RNA-Seq library preparation method was utilized. Differential gene expression analysis based on AR status was performed, yielding 8479 differentially expressed genes (DEGs) for HCCLM3 and 8152 DEGs for SNU475. Next, gene set enrichment analysis (GSEA) was performed using the v.2024.1.Hs molecular signature database (MSigDB) and hallmark gene sets (H) [35,36].

### 2.5. CRISPR AR Knockout

Multi-guide sgRNA was rehydrated (1.5 nmol with 15 µL 1xTE for a final concentration of 100 µM). Prior to RNP complex assembly, sgRNA was diluted to 30 µM with 1xTE. RNPs were prepared by mixing 18 µL supplemented Nucleofector solution (Solution SE for SNU475 and HCCLM3), 6 µL 30 µM sgRNA, and 1 µL 20 µM Cas9. Cells were centrifuged (1.5 × 10^5^ cells per transfection; 1K, 5 min); cell pellets were resuspended in 5 µL of supplemented Nucleofection solution SE and added to 25 µL of RNP solution. Cell/RNP suspensions were transferred into a Nucleocuvette strip and pulsed with program EN-150 for SNU475 or DS-120 for HCCLM3 cells. AR CRISPR-transfected cells were harvested, suspended at 250 cells in 50 mL medium, and plated out on five 96-well plates (0.1 mL/well). Individual single clones were isolated and expanded and cell lysates tested by ELISA (PathScan^®^ Total Androgen Receptor Sandwich ELISA Kit; #12850, Cell Signaling Technology, Danvers, MA, USA). Genomic DNA was isolated from AR protein-negative clones, PCR-amplified with primers 5′-CTGCTCCGCTGACCTTAAA-3′ (sense) and 5′-CTTTGGTGTAACCTCCCTTGA-3′ (anti-sense), and subjected to Sanger sequencing to confirm appropriate edits.

### 2.6. Protein Extraction and Immunoblotting

RIPA Lysis and Extraction Buffer (89900, Thermo Scientific, Waltham, MA, USA) and Halt Protease and Phosphatase Inhibitor Cocktail (78440, Thermo Scientific, Waltham, MA, USA) were used according to their respective product instructions to prepare all cell lysates. Lysates were stored at −80 °C prior to use. Protein amounts were determined by the BCA assay (23227, Thermo Scientific, Waltham, MA, USA), and the lysate protein quantity was equalized for all samples across a single assay. Samples were loaded on a 4-12% SDS-PAGE gel, and gel electrophoresis was used to resolve the samples prior to transfer. Proteins were transferred from the gel onto a nitrocellulose membrane using a Trans-Blot^®^ Turbo^™^ Transfer System (#1704271 and #1704150, Bio-Rad Laboratories, Hercules, CA, USA). Membranes were washed three times with TBST [Tris-buffered saline (TBS) containing 0.1% Tween 20] before blocking with 5% non-fat dry milk in TBST for 1 h. Following three additional wash steps, membranes were incubated overnight with a specific primary antibody in TBST (1:1000) at 4 °C. The following day, the membranes were washed three times in TBST and then incubated with either goat anti-rabbit or anti-mouse IgG-horseradish peroxidase (HRP)-conjugated secondary antibodies (1:5000) for 1 h at 25 °C. Selected proteins were detected with HRP chemiluminescence reagent. The following primary antibodies were used: anti-AR-NT (#5153) and anti-GAPDH (#5174) (Cell Signaling Technology, Beverly, MA, USA).

### 2.7. AR Protein ELISA

Androgen receptor protein levels were measured by AR ELISA (PathScan^®^ Total Androgen Receptor Sandwich ELISA Kit, #12850, Cell Signaling Technology, Danvers, MA, USA). Cells were plated at 3 × 10^5^ cells/well in 6-well plates and incubated overnight to allow for adherence (5% CO_2_, 37 °C). Cells were treated with semi-log concentrations of the test compound in 10% FBS and 0.1–1% vehicle (DMSO for Nic/Cmp#7/Cmp#11), depending on the solubility of the compound. Cells were incubated for 24 h treatment. Cell lysates were prepared with RIPA Lysis and Extraction Buffer (89900, Thermo Scientific, Waltham, MA, USA) and Halt Protease and Phosphatase Inhibitor Cocktail (78440, Thermo Scientific, Waltham, MA, USA), according to product instructions. Lysates were stored at −80 °C prior to use. Protein concentrations were determined via the Bradford assay (5000006, Bio-Rad Laboratories, Hercules, CA, USA) using a Biotek Synergy H1 plate reader. Then, 80 µg of protein was loaded per ELISA well immediately after estimation. The AR ELISA protocol was carried out according to the manufacturer’s instructions. Optical densities were measured at 450 nm using a Biotek Synergy H1 plate reader. 

### 2.8. Invasion Assay

The BioCoat Matrigel invasion assay was performed according to the manufacturer’s instructions (BioCoat Matrigel Invasion Chambers, 354480, Corning, Corning, NY, USA). AR-negative clones and parental lines (SNU475 and HCCLM3) were hormone-deprived in phenol red-free RPMI1640 with 5% csFBS (±Pen-Strep) for 24 h. A total of 40,000 cells were suspended in a well insert in 0.5 mL of phenol red-free medium without csFBS over a well of 0.75 mL phenol red-free medium with 5% csFBS as a chemoattractant and incubated for 24 h. Invaded cells were fixed in 4% paraformaldehyde and stained with 0.1% crystal violet in 20% methanol. Cells in 10 different and non-overlapping fields of each membrane were photographed and counted. The Fluorometric QCM™ 24-Well Cell Invasion Assay (ECM 554, MilliporeSigma, Burlington, MA, USA) was executed as per the manufacturer’s protocol using 1.25 × 10^5^ cells in 250 μL of serum-free medium and incubated for 24–72 h. Following cell lysis and staining with CyQuant GR Dye, fluorescence was measured in a plate reader using a 480/520 nm filter set.

### 2.9. Cytotoxicity Assay

The cytotoxicity of compounds was determined by the CCK-8 assay (Cell Counting Kit-8, CK04, Dojindo Molecular Technologies, Kumamoto, Japan). Cell lines were plated at 10,000 cells/well in 96-well plates and incubated overnight to allow for adherence (5% CO_2_, 37 °C). Cells were treated in triplicate with semi-log concentrations of the test compound in 10% FBS and 0.1–1% vehicle (DMSO for Nic/Cmp#7/Cmp#11), depending on the solubility of the compound. Cells were placed back in the incubator for 72 h treatment. CCK-8 reagent was added, and plates were further incubated 1–4 h as directed by the supplier’s instructions. Optical density was measured using a Biotek Synergy H1 plate reader at 450 nm. Primary hepatocytes were plated at 30,000 cells or 50,000 cells/well and maintained according to the technical information bulletin provided by the supplier. Upon treatment, maintenance media were supplemented with 10% FBS and fresh test compound for all subsequent media changes, as directed.

### 2.10. Pharmacokinetic Studies and Parameter Analyses

The pharmacokinetics of niclosamide, valine–niclosamide, compound #7, valine–compound #7, compound #11, and valine–compound #11 after IV and PO administration were assessed in male C57BL/6 mice. The in-life study and quantitative analysis of plasma samples using LC/MS were performed at Charles River Laboratories, Inc. (Wilmington, MA, USA). Mice (approximately 30 g, *n* = 3 per group) were administered a single dose of niclosamide, compound #7, or compound #11 at 2 mg/kg (IV) or 40 mg/kg (PO) or valine–niclosamide (IV: 2.8 mg/kg; PO: 56.4 mg/kg), valine–compound #7 (IV: 2.77 mg/kg; PO: 55.49 mg/kg), or valine–compound #11 (IV: 2.9 mg/kg; PO: 56.18 mg/kg) at molar equivalent IV and PO doses to their respective parent compounds. Blood samples were collected serially at 0.083, 0.25, 0.5, 1, 2, 6, and 12 h after IV dosing, with the exception of valine–compound #7, where collection at 1 h was omitted and collection at 24 h was performed, and at 0.25, 0.5, 1, 2, 4, 8, and 12 h after PO dosing, with the exception of compound #7 and valine–compound #7, where collection at 4 h was omitted and collection at 24 h was performed.

For pharmacokinetic studies and subsequent in vivo studies, niclosamide and its analogs were dissolved in DMSO–Cremophor EL–water (3:15:82 by volume). The selection of the DMSO–Cremophor EL-based vehicle used for all in vivo experiments resulted from the evaluation of several published formulations, culminating in the identification of this vehicle as producing a clear solution at an appropriate concentration for IV niclosamide administration in the initial PK studies [37]. In other formulations tested, niclosamide was incompletely soluble, forming slurries or suspensions (5% Methocel A4M in water; 5% Tween 80/5% ethanol in saline) [15,38], or formed a solution containing a high proportion of organic solvent lacking translational compatibility (67% PEG400/33% N,N dimethylacetamide) [39]. While Cremophor-based drug formulations for oncology applications are still available, Cremophor EL has been associated with toxicities and altered drug disposition [40]. Thus, continued pre-formulation efforts provide opportunities to potentially improve the oral tolerance and bioavailability of niclosamide derivatives.

### 2.11. Thermodynamic Solubility Study

The thermodynamic solubility of niclosamide, valine–niclosamide, compound #7, valine–compound #7, compound #11, and valine–compound #11 was assessed in simulated gastric fluid (SGF) (0.2% (*w*/*v*) sodium chloride in 0.7% (*v*/*v*) hydrochloric acid, deionized water, 0.32% pepsin (*w*/*v*), pH 1.2 ± 0.05) and in fasted-state simulated intestinal fluid (FaSSIF) (0.056% (*w*/*v*) lecithin, 0.161% (*w*/*v*) sodium taurocholate, 0.39% (*w*/*v*) monobasic potassium phosphate, 0.77% (*w*/*v*) potassium chloride, deionized water, pH 6.5 ± 0.05). The thermodynamic solubility analysis in biological media and the quantitative analysis of samples were performed according to the standard protocol at WuXi AppTec (Shanghai, China).

### 2.12. Pharmacokinetic Modeling Methods

A total of 36 mice with available PK data were included for non-linear mixed-effects analysis using NONMEM (version 7.5.0, ICON Development Solutions, Ellicott City, MD, USA), implementing the first-order conditional estimation with interaction (FOCE-I). R (version 4.2.0, R Foundation for Statistical Computing, Vienna, AUT, http://www.r-project.org) was used for visual diagnostics. PK data included in the analysis comprised the plasma concentration–time data of valine conjugates (valine–niclosamide, valine–compound #7, and valine–compound #11) and the parent compounds (niclosamide, compound #7, and compound #11), obtained from the in vivo cleavage of valine conjugates post-administration, and the plasma concentration–time data of the parent compounds (niclosamide, compound #7, and compound #11) administered to a separate cohort of animals. The analyte (valine conjugate and parent compound) plasma concentration–time data were confirmed to have a biphasic distribution and elimination and were therefore fit separately to a linear two-compartment model with a depot for oral administration. Each model was parametrized in terms of clearance (CL), the volume of distribution of the central compartment (V1), intercompartmental clearance (Q), and the volume of the peripheral compartment (V3). Interindividual variability (IIV) for each parameter was assumed to be log-normally distributed, and residual variability (RUV) was described with a proportional error model. The PK parameter estimates from these independent models (parent compound or valine conjugate) were used as initial estimates for a combined model describing the conversion and disposition of the valine conjugate to the parent drug and associated metabolites. The model estimated the fraction of pro-drug converted to parent drug (Fm), which was used to calculate the first-order rate constant of metabolite formation (Kmet) = Fm × Kel. This rate constant was used to calculate the half-life of the conversion process and the amount of metabolite formed at complete conversion. The rate of metabolite formation was then calculated as the ratio of the amount of metabolite formed to the time required for complete conversion (~5 half-lives), as detailed in Appendix A. 

### 2.13. Plasma Stability Study

The plasma stability of valacyclovir, valine–niclosamide, valine–compound #7, and valine–compound #11 was assessed in human male plasma and male C57BL/6 murine plasma with K_2_EDTA as an anti-coagulant. Samples were taken at 0, 30, 60, 120, 180, and 360 min to quantitate the disappearance of the valine conjugates and appearance of a cleaved parent. The matrix stability assay in plasma and the quantitative analysis of samples by UV-VIS HPLC were performed according to the standard protocol at Charles River Laboratories, Inc. (Wilmington, MA, USA).

### 2.14. Animal Tolerability Study

A tolerability study of valine–niclosamide was performed in male C57BL/6J mice (8 weeks old, Jackson Laboratories, Bar Harbor, ME, USA) to inform dose selection for follow-up efficacy studies in a model of HCC. Mice were group-housed under conditions of a constant photoperiod (12 h light/12 h dark), temperature, and humidity with ad libitum access to water and standard pelleted chow. Mice were randomized to treatment groups (*n* = 5 per group) that received valine–niclosamide at 14.1, 56.4, and 141.0 mg/kg, corresponding to molar equivalent niclosamide doses of 10, 40, and 100 mg/kg. Control mice were treated with the vehicle (DMSO–Cremophor EL–water, 3:15:82 by volume). Treatments were administered PO by gavage once daily for 14 days. Mouse weights were measured every 2 days. At the study endpoint, mice were euthanized approximately 2–4 h after the last dose. Blood was collected by cardiac puncture immediately after CO_2_ euthanasia. At necropsy, carcasses were examined for grossly visible lesions, and organ weights were measured (liver, kidneys, spleen, urogenital tract, testes, heart, lungs, brain). Whole blood and serum were submitted to the Comparative Pathology and Digital Imaging Shared Resource at the Ohio State Comprehensive Cancer Center (OSUCCC, Columbus, OH, USA) for the determination of complete blood counts and serum chemistry. This study was conducted according to protocols approved by the Ohio State University Institutional Animal Care and Use Committee.

### 2.15. Hollow Fiber Assay

The hollow fiber procedures were based upon those developed by Hollingshead et al. [41] with technical consultation kindly provided by Drs. Joanna Burdette and Dan Lantvit (University of Illinois Chicago, Chicago, IL, USA). We modified this technique, based upon literature support, in order to utilize the solely AR-SV-expressing SNU475 model, which has a low frequency of xenograft take upon subcutaneous injection of cells in nude mice but does show exponential growth in a hollow fiber assay (HFA) model [42]. In order to utilize a PO dosing approach with the HFA, we extended the traditional HFA timeline to establish angiogenesis, as noted in the literature [43,44]. Based upon our own cell density and angiogenesis optimization, we arrived at the following approach. Hollow fibers (KrosFlo Implant Membranes, M138615, Repligen, Waltham, MA, USA) were filled with 3 × 10^6^ SNU475 cells/mL per length and heat-sealed into 2 cm lengths on Day −1. On Day 0, three fibers were implanted into 20-week-old male outbred homozygous athymic nude mice (*Foxn1^nu^/Foxn1^nu^*, The Jackson Laboratory, Bar Harbor, ME, USA) subcutaneously parallel to the spine. Following a 14-day waiting period to allow for angiogenesis around the fibers, treatment was initiated. Mice were randomized to treatment groups (*n* = 4 per treatment group and *n* = 3 per vehicle group) and treated BID PO for 7 days with either niclosamide (75 mg/kg), valine–niclosamide (106.1 mg/kg, 75 mg/kg molar equivalent), valine–compound #7 (87.3 mg/kg, 75 mg/kg molar equivalent multiplied by a factor of 0.78 to match exposure to valine–niclosamide), compound #7 (62.9 mg/kg, 87.3 mg/kg valine–compound #7 molar equivalent), or vehicle (3% DMSO, 15% Cremophor EL in water). We performed an additional study (*n* = 3 per group, 2 fibers per mouse) of mice treated with either the vehicle (25% ethanol, 25% Cremophor EL in physiologic saline) or EOD IP paclitaxel (PTX) (25 mg/kg) as a positive control based upon efficacy shown by Mi et al. and efficacy against SNU475 cells [45,46]. On Day 21, mice were euthanized and hollow fibers were explanted. The MTT assay was performed according to the KrosFlo Implant Membrane protocol utilizing Thiazolyl Blue Tetrazolium Bromide (M2128, MilliporeSigma, Burlington, MA, USA).

### 2.16. Statistical Analyses

Statistical tests were performed in GraphPad Prism (version 10; GraphPad Software Inc., La Jolla, CA, USA) and data presented as the mean or geometric mean ± confidence interval or coefficient of variation, unless otherwise noted. Statistical significance was ascertained through an unpaired *t*-test for two groups or an analysis of variance (ANOVA) for 3 or more groups, unless otherwise noted in specific figure legends. *p* values of <0.05 were considered statistically significant.

## 3. Results

### 3.1. AR/AR-SV Regulated Oncogenic and Tumor-Suppressive Pathways

Building upon our previous reporting of AR-SV transcript presence and abundance in patients within The Cancer Genome Atlas (TCGA) Liver Hepatocellular Carcinoma (LIHC) cohort [5], we evaluated the relationship between AR-SV transcript abundance and overall survival. AR-SV transcript abundance was expressed as a percentage of AR-FL transcript abundance, and then tumors in the LIHC cohort were stratified into quartiles, with Quartile 1 (Q1) representing the lowest abundance and Quartile 4 (Q4) representing the highest, and analyzed for differences in overall survival (OS) among the quartiles. The overall survival of patients represented in Q1 was significantly higher than in patients in Q4 (Figure 1A), indicating that overall survival is significantly reduced with increased AR-SV transcript abundance.

To better understand the impact of AR-SV transcripts on downstream cellular signaling, we analyzed the effects of siRNA-mediated AR depletion on gene expression in two human HCC cell lines, HCCLM3 and SNU475, with differing AR/AR-SV expression statuses. HCCLM3 cells express both AR-FL and AR-SV, representative of the AR tumor status in most HCC patients, while SNU475 cells solely express AR-SV due to a genomic deletion in the ligand-binding domain (LBD) [5]. Upon confirmation of successful siRNA-mediated knockdown of the total AR protein compared to non-specific siRNA-transfected controls, we analyzed the resulting differentially expressed genes (DEGs) between the AR knockdown and control conditions for each cell line. AR knockdown in HCCLM3 cells resulted in 8479 DEGs, while that in SNU475 cells resulted in 8152 DEGs, with an overlap of 3701 genes common to both sets of DEGs (Figure 1B). We next used gene set enrichment analysis (GSEA) and the hallmark gene set within the Molecular Signature Database (MSigDB) to determine the correlations between the AR-dependent HCCLM3 and SNU475 gene sets and various characterized biological signaling pathways. Several pathways were significantly enriched in both gene sets, with false discovery rates (FDR) below 25%. We found that the presence of siAR downregulated several major regulatory and cell proliferation pathways in both cell lines, including E2F targets, G2/M checkpoints, Myc targets, and mitotic spindle signaling (Figure 1C,D). These pathways are all likely upregulated by AR/AR-SVs. We also found that, in the presence of siAR, TNFα signaling via NF-κB, IL-2-mediated STAT5, IL-6-mediated STAT3, and KRAS signaling were all upregulated (Figure 1C,D). This indicates that AR/AR-SVs likely play a role in suppressing these oncogenic signaling pathways.

### 3.2. CRISPR AR KO Downregulates Cellular Invasion

We decided to further validate the role of AR/AR-SVs in downstream oncogenic pathways by utilizing CRISPR-Cas9 to produce HCCLM3 and SNU475 AR knockout cell lines (Sanger sequencing confirmation shown in Appendix A). AR KO was confirmed for multiple clones for both HCCLM3 and SNU475 by comparing the total AR protein levels in edited clones to those in parental HCCLM3 or SNU475 (Figure 2A).

Based upon our GSEA data suggesting a role for AR in regulating E2F targets—transcription factors regulating the cell cycle and implicated in cancer progression, invasion, and metastasis—we examined the invasive potential of our AR KO clones. We found that the AR KO clones had decreased invasive potential compared to parental unedited cells in our invasion assays (Figure 2B).

### 3.3. Niclosamide Analogs Decrease AR Expression and Invasion in AR/AR-SV(±) HCC Cells

Based upon previous reports supporting niclosamide’s ability to decrease AR/AR-SV protein levels, as well as decrease the activity of multiple oncogenic pathways, we decided to evaluate niclosamide and two previously reported analogs with anti-AR effects, compound #7 and compound #11, for use in HCC (Figure 3A) [32]. We assayed the ability of niclosamide, compound #7, and compound #11 to decrease the AR protein in two human HCC cell lines, SNU423 and SNU475, with differing AR profiles. SNU423 is a human HCC cell line expressing only AR-FL, while SNU475, as previously mentioned, only expresses AR-SVs [5]. Niclosamide was effective in decreasing AR protein levels after 48 h of treatment time in SNU423, with an IC_50_ of roughly 0.2 µM, and in decreasing AR-SV protein levels in SNU475, with an IC_50_ of roughly 0.4 µM (Figure 3B). Compound #7 and compound #11 were also effective in decreasing AR-FL in SNU423, with IC_50_ values of 1.0 µM and 1.1 µM, respectively, and AR-SV in SNU475, with IC_50_ values of 4.6 µM and 3.4 µM, respectively (Figure 3C,D). We also compared niclosamide, compound #7, and compound #11 to both the AR antagonist enzalutamide and the kinase inhibitor standard-of-care HCC agent sorafenib at the clinically relevant concentration of 10 µM [47,48]. We found that all three niclosamide-based compounds were able to more effectively decrease AR protein levels compared to enzalutamide and sorafenib (Appendix A).

Additionally, we tested the ability of niclosamide to decrease SNU475 invasion, as we did for the SNU475 CRISPR AR KO cell lines, and found decreased invasion after 48 h of treatment at 1 µM and 10 µM (Appendix A). Critically, cell viability at 48 h was not impacted at 1 µM, supporting niclosamide’s ability to mitigate SNU-475 cellular invasion at concentrations less than 10 µM without overt cytotoxicity.

Finally, we determined niclosamide’s anti-proliferative impact on SNU423 and SNU475 along with HepG2, a human HCC cell line lacking AR expression; THLE-2, an AR-negative immortalized normal human liver cell line; and primary male hepatocytes (Table 1, Appendix A) [5]. Niclosamide demonstrated some cancer selectivity, with greater potency in inhibiting SNU423 and SNU475 cell growth after 72 h of treatment time (IC_50_, 0.25 and 1.3 µM, respectively) as compared to primary male hepatocytes (7.2 µM), while similar anti-proliferative potency was apparent in HepG2 and THLE-2 (0.6 and 0.37 µM, respectively). Compound #7 and compound #11 showed similar potency to niclosamide against SNU423 cells (0.28 and 0.58 µM, respectively) but were less potent against SNU475 cells (4.0 and 7.2 µM, respectively). Compound #7 and compound #11 showed anti-proliferative potencies in a similar range to that of niclosamide against both HepG2 (0.45 µM and 1.9 µM, respectively) and THLE-2 (0.31 µM and 0.56 µM, respectively). Like niclosamide, compound #7 and compound #11 demonstrated some cancer selectivity, with lower potency against primary male hepatocytes (>30 µM and 9.5 µM, respectively). Compared to enzalutamide, a prostate cancer therapeutic, and sorafenib, lenvatinib, and regorafenib, which are standard-of-care therapeutics in liver cancer [49], niclosamide and its analogs showed a higher degree of potency against AR-FL(±) SNU423 HCC cells, while only sorafenib and regorafenib had similar degrees of potency against AR-SV(±) SNU475 HCC cells at 4.0 µM and 1.6 µM, respectively (Appendix A).

### 3.4. Niclosamide Analogs Demonstrate Improved Pharamcokinetics and Solubility

Based upon the prior reports of improved rat pharmacokinetic parameters with compound #7 compared to niclosamide [32], we assessed the pharmacokinetics of both compounds #7 and #11 in single-dose IV and PO studies in mice (Figure 4A and Appendix A). Consistent with limiting metabolism, both compound #7 and compound #11 showed significant decreases in systemic clearance following an IV dose compared to niclosamide (0.04 and 0.03 L/h, respectively, versus 0.08 L/h, Appendix A). These apparent reductions in systemic clearance were matched with improvements in oral exposure such that compounds #7 and #11 exhibited significant increases in circulating drug levels following an oral dose compared to niclosamide (12.97 and 7.98 h × µmol/L AUC_all_, respectively, versus 1.79 h × µmol/L). While none of the niclosamide analogs tested exhibited solubility above 1.5 µM in simulated gastric fluid, compounds #7 and #11 showed increased solubility in simulated intestinal fluid compared to niclosamide (Figure 4B).

### 3.5. Valine–Niclosamide and Valine-Conjugated Niclosamide Analogs Demonstrate Improved Oral Exposure, Solubility, and Bioavailability

With the goal of sustaining roughly 5 µM plasma levels for 8 h of a daily dosing interval to achieve anti-HCC effects in subsequent xenograft models, we linearly scaled our pharmacokinetic data and approximated the need for 500–1500 mg/kg doses of niclosamide and its analogs. Despite the apparent improvements in pharmacokinetics provided by compounds #7 and #11, the necessary dose size remained prohibitively large. To address this limitation, we pursued an approach to improve on a known limitation of the niclosamide pharmacophore—poor solubility [13]. To this end, we further modified niclosamide, compound #7, and compound #11 by valine amino acid conjugation (Figure 5A, Appendix A) [50] to determine if a pro-drug approach could improve the solubility and subsequent exposure following an oral dose. We determined the impact of this structural modification on pharmacokinetic parameters using single-dose IV and PO pharmacokinetic studies of the valine-conjugated compounds in mice (Figure 5B), which revealed that the valine-conjugated molecules had decreased systemic IV clearance compared to the parent compounds, with significant reductions for both niclosamide and compound #7 (0.02 versus 0.08 L/h and 0.01 versus 0.04 L/h, respectively). Additionally, conjugation resulted in substantial increases in the systemic exposure of pro-drugs following an equimolar oral dose of valine–niclosamide and valine–compound #7 compared to a 40 mg/kg dose of their respective parental compounds (48.57 versus 3.25 h × µmol/L and 62.09 versus 3.58 h × µmol/L, respectively; Appendix A and Appendix A). We also observed the significantly higher oral bioavailability of valine–niclosamide compared to niclosamide, although this was not observed with valine–compound #7 compared to unconjugated compound #7 (Appendix A).

While valine–compound #11 had the lowest IV clearance of all the conjugated compounds (0.004 L/h), it had the lowest systemic exposure of all valine-conjugated compounds following an oral dose (4.17 h × µmol/L), similar to that of unconjugated niclosamide (3.25 h × µmol/L) (Figure 5B). This was particularly striking as valine–compound #11 and valine–compound #7 differed by a single chlorine atom but showed dramatic discrepancies in the exposure of their pro-drug forms following equivalent oral doses. To determine the potential impact of solubility on valine conjugate pharmacokinetics, we conducted thermodynamic solubility studies in simulated gastric and intestinal fluids (Figure 5C). In simulated gastric fluid, the valine-conjugated forms showed 9.2 to 17.7 µM solubility, a notable improvement over the extremely limited solubility of the three unconjugated analytes (all less than the 1.56 µM LLOQ). However, in simulated intestinal fluid, while all other niclosamide analogs and their respective valine-conjugated forms showed at least a two-fold increase in solubility compared to niclosamide (Figure 4B versus Figure 5C), valine–compound #11’s solubility remained under 1.56 µM LLOQ in intestinal fluid, providing a potential explanation for its reduced systemic exposure following an oral dose relative to other analogs.

### 3.6. Valine–Niclosamide in an In Vivo Hollow Fiber Assay Model of AR-SV(±) HCC

Based upon our pharmacokinetic data showing the improved systemic exposure of valine–niclosamide and valine–compound #7 following an oral dose, we decided to move forward with valine–niclosamide and valine–compound #7 for study in vivo. To assess the in vivo efficacy of these compounds against AR-SV(±) liver cancer, we utilized a modified hollow fiber assay (HFA), as developed by Hollingshead et al., to accommodate the poor subcutaneous tumor take rate of SNU475, our primary AR-SV (±) HCC model of interest [41]. We utilized paclitaxel as a positive control due to its prior use as a positive control in HFA models [45].

To inform our dose selection, we performed a two-week tolerability study of valine–niclosamide in mice administered daily oral doses of 10, 40, and 100 mg/kg molar equivalencies of niclosamide (Appendix A). Based upon valine–niclosamide’s increased oral exposure, we wanted to determine whether higher oral exposure led to potentially dose-limiting toxicities. We found that valine–niclosamide was tolerated at all tested dose levels, with no significant impact on body weight or liver enzyme levels or concerning increases in spleen weight.

We designed our efficacy study to target 1 to 4 µM circulating levels of agent for 8 h of our planned 12 h BID dosing interval, given that the previously determined IC_50_ values of niclosamide and compound #7 against SNU475 were 1.3 µM and 4.0 µM, respectively. Based on linearly scaling our pharmacokinetic data, 75 mg/kg niclosamide BID and a molar equivalent dose of valine–niclosamide (105.7 mg/kg BID) were expected to provide plasma levels of approximately 0.2 µM niclosamide and 3.5 µM valine–niclosamide for up to 8 h post-dose. In an effort to predict the tolerated doses of compound #7 and its valine conjugate, we then calculated a dose for valine–compound #7 that was based on matching the systemic exposure to a 105.7 mg/kg valine–niclosamide dose. Using this pharmacokinetic rationale, we reasoned that 87.3 mg/kg BID valine–compound #7 should not be overtly toxic and then determined a subsequent equimolar dose of compound #7 (62.9 mg/kg BID) to deliver as a comparator. Based on our pharmacokinetic data, these dosing regimens were expected to provide plasma levels of 3.7 µM valine–compound #7 and 0.86 µM compound #7 up to 8 h post-dose. To ensure that our subcutaneous fibers were sufficiently vascularized to receive circulating therapy, we performed a positive control experiment by treating mice with 25 mg/kg paclitaxel (PTX) IP every other day for one week (Figure 6A). PTX treatment resulted in a significant decrease in cell viability as determined by the MTT assay within our fibers, confirming the validity of our model system. In contrast to our control experiment, one week of BID treatment with niclosamide, compound #7, valine–niclosamide, and valine–compound #7 resulted in no differences in cell viability within the fibers (Figure 6B). Of note, one mouse was removed early from the valine–niclosamide group due to potential gavage-associated injury, and two mice were excluded from the valine–compound #7 group due to apparent toxicity.

### 3.7. Assessing Conversion of Valine-Conjugated Niclosamide Analogs to Parent

To better understand the lack of efficacy in our hollow fiber assay, we revisited the cleavage of valine–niclosamide and valine–compound #7 to their parental forms as incomplete conversion could have limited the amount of active agent reaching our SNU-475 cells (Appendix A, Appendix A). When measuring circulating niclosamide following a 56.4 mg/kg oral dose of valine–niclosamide (molar equivalent of 40 mg/kg niclosamide), we found a niclosamide C_max_ of only 0.87 µM, which was less than the 1.65 µM C_max_ resulting from simply dosing 40 mg/kg niclosamide (Appendix A). Likewise, the AUC_all_ of niclosamide following an oral dose of valine–niclosamide was only 0.8 µM × h, which was lower than the AUC_all_ of 3.25 µM × h following an equivalent dose of unconjugated niclosamide. Compound #7 followed a similar pattern in that its C_max_ following a 55.49 mg/kg oral dose of valine–compound #7 was limited to 0.85 µM, which was lower than the C_max_ of 5.78 µM resulting from an equivalent dose of compound #7. The AUC_all_ of compound #7 following an oral dose of valine–compound#7 was only 1.31 µM × h, which was lower than the AUC_all_ of 12.97 µM × h following an equivalent dose of unconjugated compound #7.

We further evaluated the conversion of valine-conjugated pro-drugs utilizing a pharmacokinetic modeling approach whereby the kinetics of both the pro-drug and parent could be simultaneously considered (Figure 7A and Appendix A). We determined that two-compartment structural models best described the combined data and allowed us to estimate the fraction of pro-drug successfully converted to the parent (Figure 7B). The estimated fraction converted was limited to 23% for valine–niclosamide, 25% for valine–compound #7, and 39% for valine–compound #11. Operating with the hypothesis that soluble esterases were at least partially responsible for cleaving our pro-drugs, as has been suggested for other amino acid conjugates [51], we performed a plasma stability assay comparing the valine-conjugated niclosamide analogs to valacyclovir, a well-characterized valine pro-drug [52,53]. Over 6 h, we found that, compared to valacyclovir, where 66.4% and 37.7% of the pro-drug remained intact in human and murine plasma, respectively, 90% or more of the valine-conjugated niclosamide analogs remained in human and murine plasma (Figure 7C). The formation of the cleaved parent was simultaneously monitored to confirm that disappearance was coupled with the production of the active parent, revealing readily detectable acyclovir but no niclosamide, compound #7, or compound #11 (Figure 7D).

## 4. Discussion

As a follow-up to our previous work characterizing the presence and role of AR-SVs in HCC sexual dimorphism, we sought to better understand downstream AR/AR-SV signaling and evaluate niclosamide, an anthelmintic drug that was recently the focus of repurposing efforts in AR-SV-driven PCa, for use as a therapeutic tool for AR-SV(±) HCC [5,32]. GSEA of AR knockdown in HCC cell lines found that AR/AR-SVs play a role in upregulating several cell cycle regulation and cellular proliferation pathways in HCC, including E2F targets, G2/M checkpoints, Myc targets, and mitotic spindle signaling. These pathways have been implicated in increasing HCC aggressiveness and proliferation [54,55]. However, several oncogenic pathways were suppressed by AR/AR-SV expression, including TNFα signaling via NF-κB, IL-2-mediated STAT5, IL-6-mediated STAT3, and KRAS signaling (Figure 1A–D). We also further solidified the role of AR in cellular invasion in AR(±) HCC by showing that AR KO was able to mitigate the cellular invasive potential (Figure 2B). As AR has previously been reported to have dual roles in HCC [4], our findings corroborate reports that AR can both contribute to cancer aggressiveness and suppress several oncogenic pathways. In this complex case, identifying a therapeutic that not only addresses AR signaling but can also suppress rebound signaling that may result from diminished AR activity is key. Niclosamide, due to its ability to inhibit a wide range of oncogenic pathways, has been previously reported as a multitool compound for cancer or other disease indications [10,11,12,13]. Niclosamide has already been investigated as an anti-AR-SV therapeutic in prostate cancer [15,28,32,33]. In addition, niclosamide is reported to inhibit the IL-6-mediated phosphorylation of STAT3, mitigate the STAT3-AR axis, and inhibit p-STAT3 binding to the PD-L1 promoter, allowing niclosamide to enhance PD-1/PD-L1 blockade [21,22,25]. Niclosamide’s effects also include inhibiting NF-kB activation and KRAS, as well as reducing cell invasion [12,20,24]. This overlap between AR/AR-SV-suppressed oncogenic pathways in AR-SV(±) HCC and niclosamide’s anti-AR and anti-cancer activity well positions niclosamide as a potential HCC therapeutic that can address this complex signaling. Additionally, with the recent approval of combination atezolizumab (PD-L1 inhibitor) and bevacizumab (VEGF inhibitor) and combination durvalumab (PD-L1 inhibitor) and tremelimumab (CTLA-4 inhibitor) as front-line therapeutics in HCC, niclosamide has strong potential as a combination therapeutic improving ICI efficacy.

Prior attempts to improve niclosamide’s therapeutic profile included a niclosamide ethanolamine salt (NEN), the development of niclosamide analogs, and a self-microemulsion of niclosamide (Nic-SMEDDS) [23,29,30,32,33]. Current research on the niclosamide pharmacophore indicates that the nitro group on the B ring and the hydroxyl group on the A ring (Figure 3A) are critical for anti-cancer activity; however, when the nitro group is substituted with a tri-fluoromethyl group, as in compound #7 and compound #11, activity is maintained and this may limit first-pass metabolism [33]. Kang et al. also reported that the A-ring chlorine is non-critical for activity. We found that the B-ring chlorine, while not critical for activity (Table 1), is critical for solubility and maintaining drug-like properties. Prior studies on similar analogs, including compounds #7 and #11, showed clear activity against AR/AR-SVs in PCa, Wnt-ß-catenin, mTORC1, and STAT3 in ovarian cancer and NF-κB in breast cancer [23,29,30,32].

While niclosamide and niclosamide analogs have shown anti-AR/AR-SV activity in PCa, this has not yet been reported in HCC [32]. In our experiments, we show the potential benefits of AR knock down on HCC cell signaling (Figure 1B–D) and show that AR knockout reduces the invasive potential of AR(±) HCC cells (Figure 2); we then provide evidence that niclosamide can reduce AR protein levels (Figure 3). These findings provide a rationale for targeting AR/AR-SVs in HCC with an agent like niclosamide, and, consistent with its effects on the AR protein, niclosamide decreased the invasion of AR-SV(±) SNU475 cells (Appendix A). Beyond its anti-invasive activity, niclosamide demonstrated favorable activity in AR(±) HCC cells in comparison to several front-line advanced HCC therapeutics, namely sorafenib, lenvatinib, and regorafenib (Table 1). Niclosamide analogs also showed either greater than or similar potency to front-line HCC agents in our in vitro cytotoxicity assays. Notably, these analogs also show improvements in mouse pharmacokinetics, including reduced systemic clearance following an IV dose and increased systemic exposure following an oral dose (Figure 3B). While these experiments validated the translatability of niclosamide activity in PCa to AR(±) HCC, prior clinical failures driven by poor solubility and absorption called for the further redesign of niclosamide to improve its bioavailability in order to leverage these effects for therapeutic use [28].

To further improve niclosamide’s bioavailability, we adopted an amino acid conjugation strategy based upon prior success in this approach with drugs like acyclovir and its pro-drug valacyclovir [52]. Although several hydrophobic amino acids have demonstrated suitability for conjugation, we first explored valine given its use in approved therapies [51]. The previously reported activity of analogs #7 and #11 provided the rationale to move forward with valine-conjugated versions of these agents [23,29,32]. Compound #31, the other lead compound presented by Liu et al., was excluded due to its structural incompatibility with valine conjugation [32]. Importantly, valine conjugation of niclosamide led to a greater increase in systemic exposure following an oral dose when compared to similar efforts. Chen et al. noted that they observed plasma concentrations of roughly 0.10 µM niclosamide 4 h after the oral administration of a 40 mg/kg dose, and NEN improved the plasma niclosamide concentration more than seven-fold to roughly 0.72 µM [14]. We similarly observed niclosamide levels of 0.22 µM at 4 h following the oral administration of a 40 mg/kg dose but found that valine conjugation increased the plasma levels more than 16-fold to 3.59 µmol/L at 4 h after a molar equivalent dose was given orally (Appendix A). The Nic-SMEDDS offered a 1.8-fold Cmax improvement over orally administered niclosamide [31]. However, we found that valine–niclosamide provided a 6.8-fold improvement in the C_max_ over orally administered niclosamide and a 14.9-fold increase in systemic exposure (Appendix A). Our thermodynamic solubility experiments showed clear improvements in solubility resulting from valine conjugation in simulated gastric fluid and, in some cases, simulated intestinal fluid (Figure 5), providing an explanation for their improved absorption. However, our experiments were not capable of separating the potential contributions of improved drug transport, as has been shown for valacyclovir via PepT1 [56], from improvements in solubility. When taken together, we demonstrated that conjugating niclosamide analogs with valine improved both solubility and absorption beyond existing approaches, but the conversion of these pro-drug forms remained problematic.

Given the poor subcutaneous tumor take rate of our cellular model of choice—SNU-475 cells, which express only AR-SVs—we were limited to an adapted HFA model as opposed to a traditional subcutaneous xenograft model. Although our model responded to PTX treatment, we were ultimately unable to demonstrate efficacy with valine–niclosamide and valine–compound #7 (Figure 6B). We hypothesized that this was potentially due to the poor conversion of our valine-conjugated pro-drugs and found that our estimated conversion rates of 0.1–0.2 µmol/h were likely insufficient to generate enough circulating active agent to generate anti-cancer effects. Consistent with this finding, our estimated fractions converted were limited to 23% and 25% for niclosamide and compound #7, respectively (Figure 7B). Our valine-conjugated pro-drugs deviated in a surprising way from other valine-conjugated pharmacophores, such as valacyclovir, which exhibits rapid and nearly total conversion to the parent in vivo [53]. As a more direct measure of activation in plasma, our plasma stability assay revealed that the amino acid-conjugated forms were essentially stable in human and mouse plasma following 6 h incubation, in stark contrast to the positive control valacyclovir, which readily produced its parent acyclovir under the same conditions (Figure 7C,D). The underlying cause of this poor cleavage remains unclear, but the additional exploration of other amino acid conjugates is warranted. Given the improvements in solubility and absorption evident following valine conjugation, improvements in the rates of bioactivation could result in therapeutic levels of circulating niclosamide or niclosamide analogs.

## 5. Conclusions

Overall, our findings further demonstrate the clinical relevance of AR/AR-SVs to HCC and their role in upregulating pro-oncogenic pathways such as E2F targets, G2/M checkpoints, Myc, and mitotic spindles. Compared to current HCC therapies that offer only minimal improvements in overall survival, niclosamide offers promise as a cancer multitool compound due to its ability to decrease the AR/AR-SV protein and prevent rebound oncogenic signaling through its activity against NF-κB, STAT3, and KRAS. In response to the clinical hurdle of niclosamide’s poor solubility and bioavailability, we present valine-conjugated niclosamide as a potentially improved therapeutic with enhanced solubility and absorption when orally administered. By potentially increasing the systemic exposure possible following an oral dose, these improvements provide a rationale for further investigation into broad applications of niclosamide to cancer or other disease indications.

There are several limitations of this work that provide opportunities for further exploration. As the roles and nature of specific AR variants have yet to be characterized in HCC as they have been in prostate cancer, this work limited its scope to AR-SVs broadly. However, the characterization of which AR-SVs are key drivers in HCC, the determination of the presence of any HCC specific variants, and the assessment of the AR variant makeup of HCC patients are key steps to further understand the role of AR-SVs in HCC as compared to PCa. As valine–niclosamide failed to significantly reduce tumor cell viability in the HFA model, likely due to poor conversion to the parent, this provides a clear opportunity to improve our understanding of valine–niclosamide’s cleavage to niclosamide and how it differs from similar pharmacophores such as valacyclovir. Due to this poor bioactivation, investigating methods to either optimize cleavage, such as alternative amino acid conjugates or enzymatic triggers, or improve the pharmacophore design, such as optimizing the linker chemistry or implementing esterase-targeted modifications, is a key next step to improving the utility of valine–niclosamide as a cancer therapeutic. Based upon the preliminary investigation of valine–niclosamide as a PepT1 substrate, additional characterization and exploration of the absorption of valine-conjugated niclosamide or other amino acid niclosamide conjugates is also warranted [57]. Additionally, while this study demonstrated links between AR-SVs and several oncogenic pathways, including E2F and Myc, further work is necessary to demonstrate whether prior reports of niclosamide’s impacts on these pathways are AR-SV-mediated. Further, the exploration of valine–niclosamide’s activity against NF-kB, STAT3, KRAS, and other oncogenic pathways that demonstrate rebound activity following AR-KO is necessary in HCC, PCa, and other cancers to demonstrate the maintenance of key types of activity across cancer types. Finally, the investigation of valine–niclosamide’s ability to enhance PD-1/PD-L1 blockade, as has been previously demonstrated with niclosamide, would offer insight into the potential for ICI combination therapy. These future directions would further solidify the potential of valine-conjugated niclosamide analogs as an anti-cancer multitool, either alone or in combination with current ICI therapeutics.

## Figures and Tables

**Figure 1 cancers-17-02535-f001:**
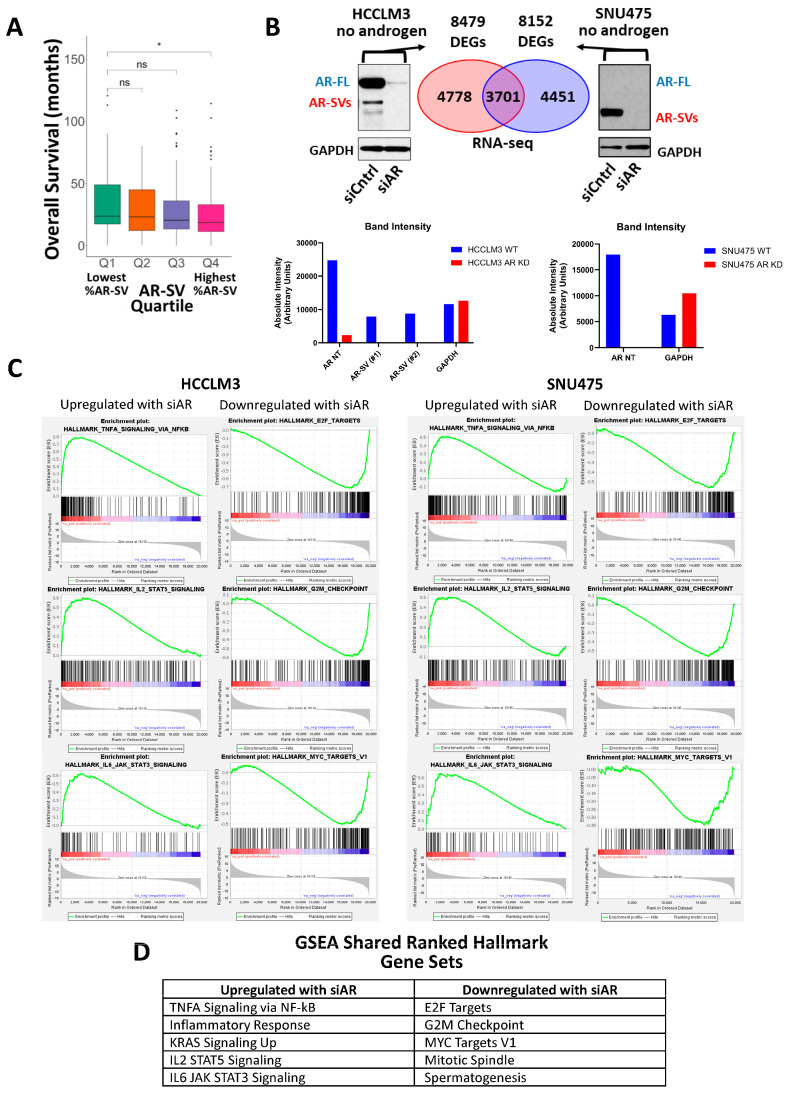
Importance and role of AR-SV signaling in LIHC primary samples and HCC cell lines. (**A**) AR-SV expression of patients in the LIHC TCGA cohort (*n* = 372), broken down into four quartiles. Statistical significance was evaluated using the Mann–Whitney test. *, *p* < 0.05 for Q1 vs. Q4. (**B**) (Top) HCCLM3 and SNU475 cells were transfected with either a non-specific siRNA control or an siRNA targeting both AR-FL and AR-SVs. The AR status of siAR-transfected cells was confirmed by Western blot with anti-AR-NT (CS#5153, Cell Signaling) and GAPDH (CS#5174, Cell Signaling). RNA sequencing revealed 8479 differentially expressed genes in the HCCLM3 line and 8152 differentially expressed genes in the SNU475 line, with an intersection of 3701 genes. (Lower Left) Band intensities of HCCLM3 WT and AR KD Western blots. AR-SV #1 indicates the top AR-SV band and AR-SV #2 indicates the bottom AR-SV band. (Lower Right) Band intensities of SNU475 WT and ARKD Western blots. (**C**) GSEA of HCCLM3 and SNU475 ranked gene lists featuring intersecting up- and downregulated hallmark gene sets. FDR < 25%. (**D**) Highest-ranking up- and downregulated intersecting hallmark gene sets from GSEA on HCC cell lines. The uncropped bolts are shown in Appendix A.

**Figure 2 cancers-17-02535-f002:**
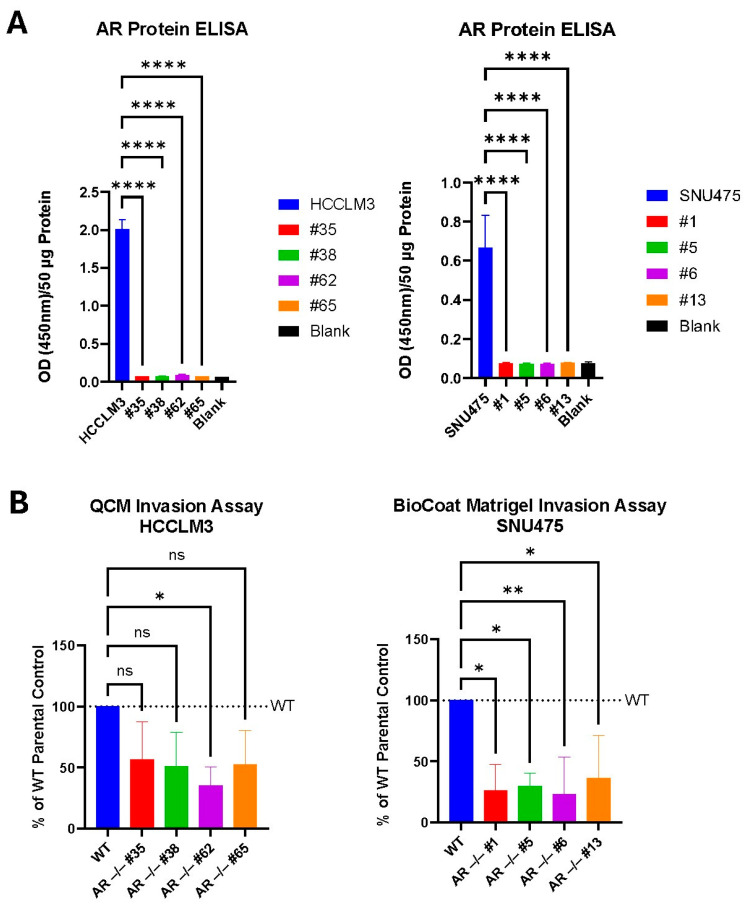
Characterization of CRISPR AR KO HCC cell lines. (**A**) AR-FL- and AR-SV-expressing HCCLM3 (left) and AR-SV-expressing SNU475 (right) CRISPR AR KO clones were generated and validated by AR ELISA. (**B**) Cellular invasion assay was performed on HCCLM3 (left) and SNU475 (right) AR KO clones. Mean ± SD. One-way ANOVA with Dunnett’s multiple comparisons test. ns, *p* > 0.05; *, *p* < 0.05; **, *p* < 0.01; **** *p* < 0.0001 versus wild type (**A**,**B**).

**Figure 3 cancers-17-02535-f003:**
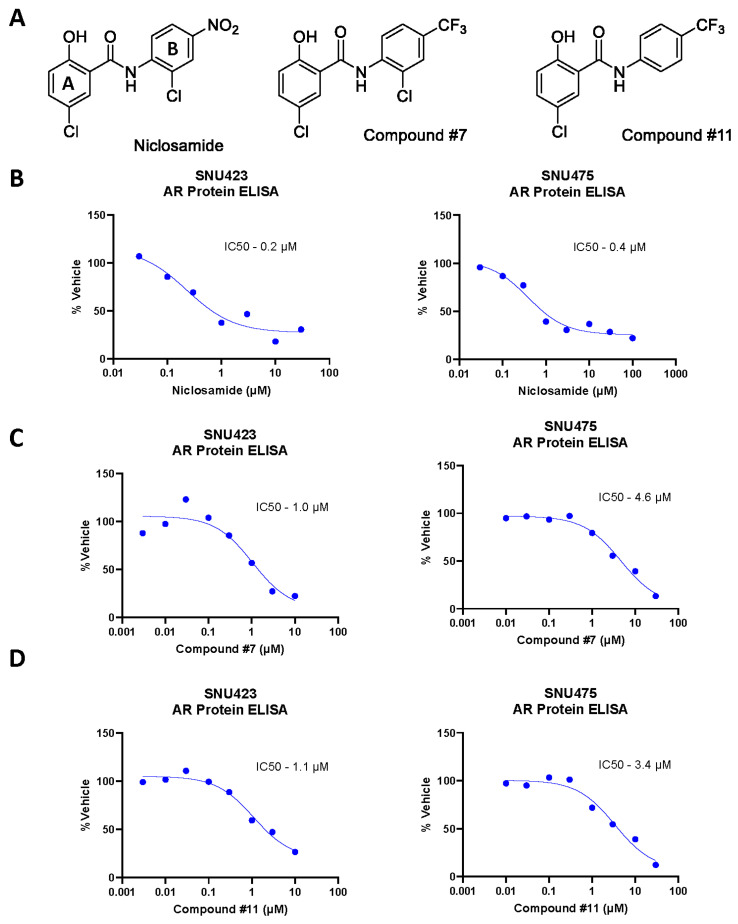
Biological activity of niclosamide analogs. (**A**) Chemical structures of niclosamide analogs. Niclosamide is labeled with A and B rings to denote typical structural reference conventions. (**B**) Niclosamide activity against AR protein determined by dose–response AR ELISA in AR-FL-expressing SNU423 (**left**) and AR-SV-expressing SNU475 (**right**) HCC cell lines. (**C**) Compound #7 activity against AR determined by dose–response AR ELISA in AR-FL-expressing SNU423 (**left**) and AR-SV-expressing SNU475 (**right**) HCC cell lines. (**D**) Compound #11 activity against AR determined by dose–response AR ELISA in AR-FL-expressing SNU423 (**left**) and AR-SV-expressing SNU475 (**right**) HCC cell lines. Curve fit and IC_50_ determined by three-parameter non-linear regression fit by least squares (**B**–**D**).

**Figure 4 cancers-17-02535-f004:**
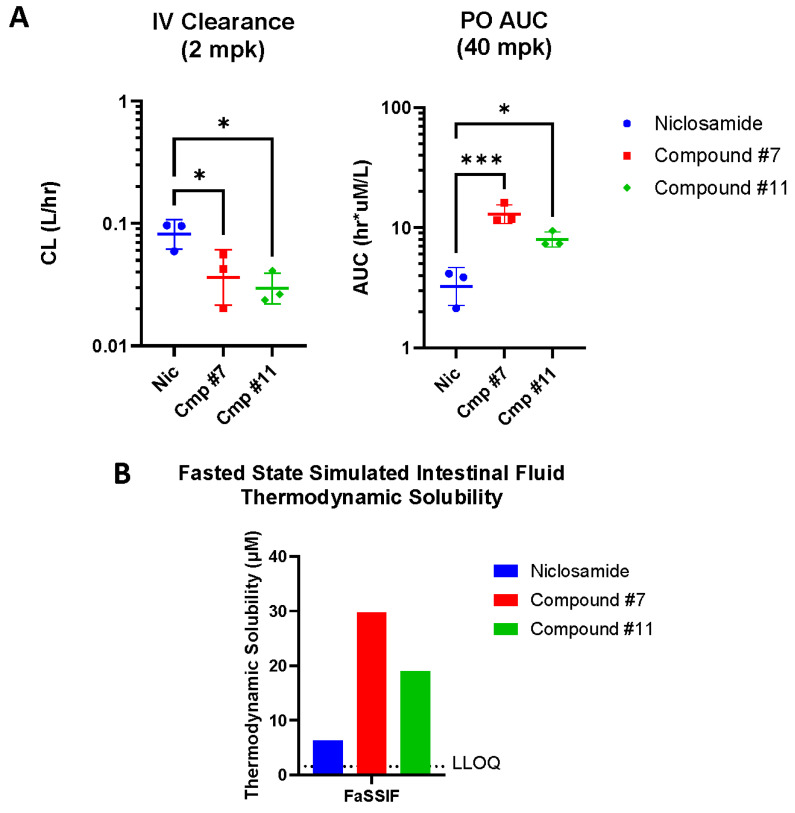
Pharmacokinetic parameters and solubility of niclosamide analogs. (**A**) Single-dose IV clearance (**left**) and single-dose PO AUC_all_ for niclosamide analogs. Geometric mean ± SD, one-way ANOVA with Šidák’s multiple comparisons test. *, *p* < 0.05; ***, *p* < 0.001. (**B**) Solubility of niclosamide analogs in fasted-state simulated intestinal fluid (FaSSIF). Solubility of niclosamide analogs in simulated gastric fluid (SGF) not shown as all values were below the LLOQ. Lower limit of quantification (LLOQ) < 1.56 µM.

**Figure 5 cancers-17-02535-f005:**
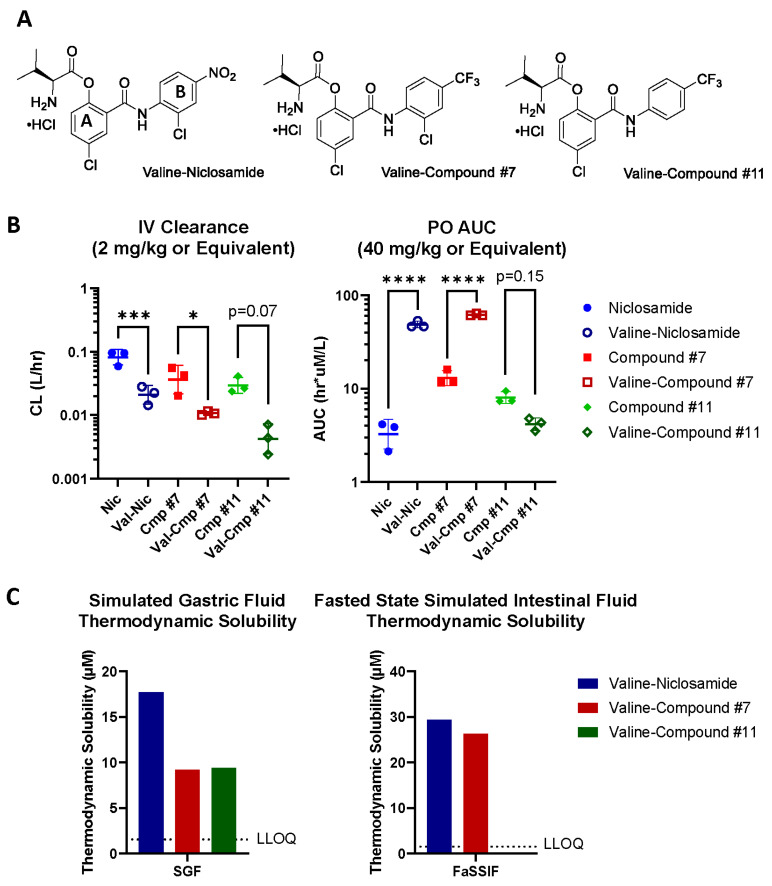
Pharmacokinetic parameters and solubility of valine-conjugated niclosamide analogs. (**A**) Chemical structures of valine-conjugated niclosamide analogs. Valine–niclosamide is labeled with A and B rings to denote typical structural reference conventions. (**B**) Single-dose IV clearance (**left**) and single-dose PO AUC_all_ for niclosamide analogs and their respective valine conjugates. Niclosamide analogs given at 2 mg/kg IV or 40 mg/kg PO with molar equivalent doses for respective valine conjugates. Geometric mean ± SD. One-way ANOVA with Šidák’s multiple comparisons test. ns, *p* > 0.05; *, *p* < 0.05; ***, *p* < 0.001; **** *p* < 0.0001. (**C**) Solubility of valine-conjugated niclosamide analogs in simulated gastric fluid (SGF) (**left**) and fasted-state simulated intestinal fluid (FaSSIF) (**right**). Lower limit of quantification (LLOQ) < 1.56 µM.

**Figure 6 cancers-17-02535-f006:**
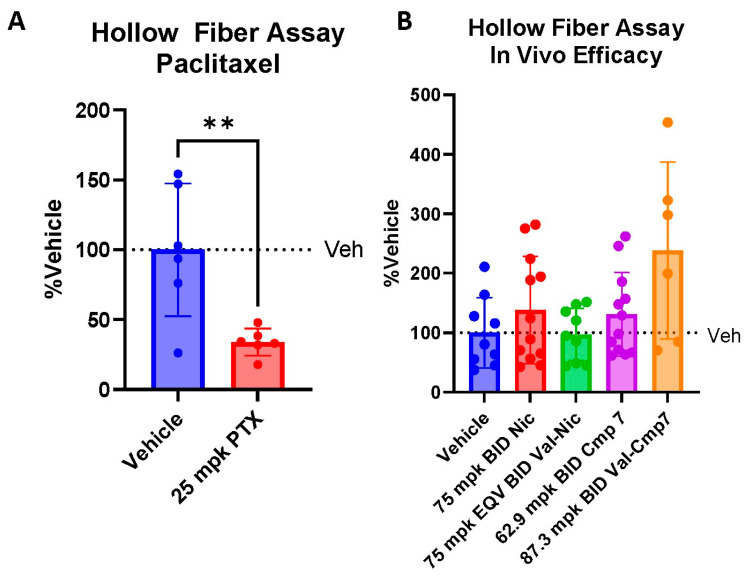
Hollow fiber assay (HFA) as a model of HCC in vivo efficacy. (**A**) Hollow fibers were filled with SNU475 cells (3 × 10^6^ cells/mL). Three nude mice were implanted with two fibers each, implanted subcutaneously on the left flank. After a two-week growth period, mice were treated with either vehicle or 25 mg/kg IP paclitaxel every other day for one week. Fibers were explanted and MTT assay was performed to determine cell viability. Mean ± SD, unpaired *t*-test. **, *p* < 0.01. (**B**) HFA performed as in (**A**). After a two-week growth period, mice were treated with either vehicle or niclosamide (75 mg/kg), valine–niclosamide (106.1 mg/kg, 75 mg/kg molar equivalent), valine–compound #7 (87.3 mg/kg, 75 mg/kg molar equivalents multiplied by a factor of 0.78 to match exposure to valine–niclosamide), or compound #7 (62.9 mg/kg, 87.3 mg/kg valine–compound #7 molar equivalents) twice daily for one week. Fibers were explanted and MTT assay was performed to determine cell viability. One-way ANOVA detected no significant reductions in cell viability.

**Figure 7 cancers-17-02535-f007:**
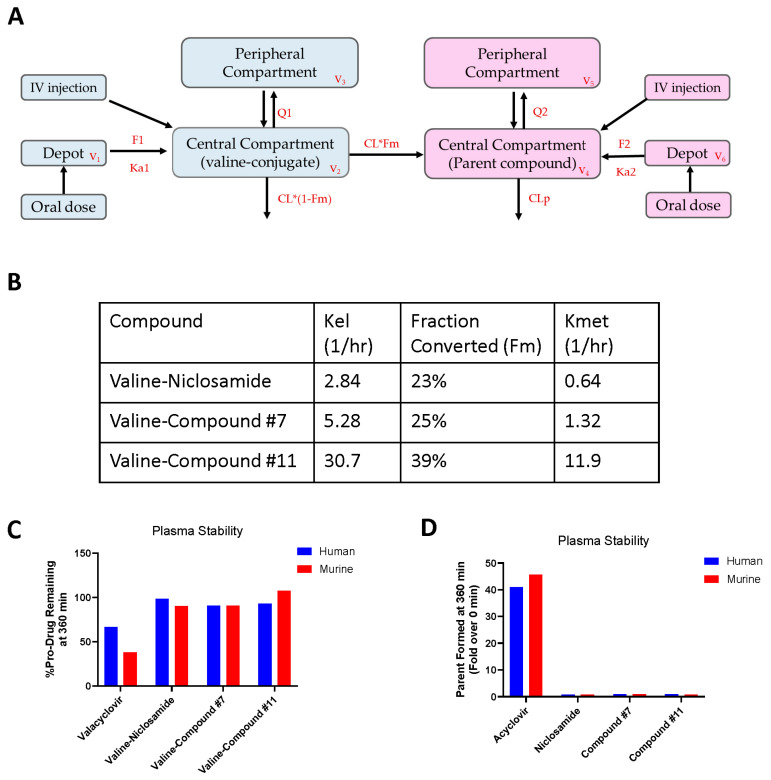
Conversion of valine-conjugated niclosamide analogs to parent drug. (**A**) Flow chart of pharmacokinetic modeling scheme integrating single IV and PO doses of both parent and valine-conjugated niclosamide. F1, bioavailability of valine conjugate; Ka1, first-order absorption rate constant of valine conjugate; V1, volume of depot for valine conjugate; Q1, intercompartmental clearance; V2, volume of central compartment of valine conjugate; CL, clearance of valine conjugate; Fm, fraction of conjugate metabolized; CLp, clearance of parent compound; V3, volume of central compartment of valine conjugate; V4, volume of peripheral compartment of parent compound; Q2, intercompartmental clearance; V5, volume of peripheral compartment of parent compound; F2, bioavailability of parent compound; Ka2, first-order absorption rate constant of parent compound; V6, volume of depot for parent compound. (**B**) Calculated rate constants and fractions converted of valine pro-drug to parent forms. Methodology outlined in Methods. Kel, elimination rate constant of valine conjugate; Fm, fraction of valine conjugate converted to parent compound; Kmet, rate constant of conversion to parent compound. (**C**) Percentages of valine-conjugated niclosamide analogs remaining in human and murine plasma following 360 min, compared to valacyclovir as a control. (**D**) Formation of parent from pro-drugs in (**C**) at 360 min, represented as fold over time 0.

**Table 1 cancers-17-02535-t001:** IC_50_ values of niclosamide and analogs against HCC, liver epithelial, and hepatocyte cell lines. ^1^ Supporting cytotoxicity data can be found in Appendix A.

IC_50_ in µM										
Compound	SNU423		SNU475		HepG2		THLE-2		Primary Male Hepatocytes	
	IC_50_	95% CI	IC_50_	95% CI	IC_50_	95% CI	IC_50_	95% CI	IC_50_	95% CI
Niclosamide	0.25	0.20–0.32	1.33	0.96–1.9	0.60	0.45–0.81	0.37	0.29–0.48	7.16	3.3–16.3
Compound #7	0.28	0.24–0.34	6.6	3.1–13.9	0.45	0.32–0.63	0.31	0.25–0.40	>30	ND
Compound #11	0.58	0.44–0.76	0.89	0.62–1.3	1.86	1.3–2.8	0.56	0.16–2.2	9.47	2.6–37.8
Enzalutamide	16.6	12.1–22.9	14.9	9.0–24.6	>30	ND ^2^	>30	ND	>30	ND
Sorafenib	6.1	4.6–8.1	4.20	2.8–6.4	10.76	6.8–16.9	>30	ND	5.10	1.2–20.4
Lenvatinib	>30	24.1–50.9	13.0	4.4–23.1	>30	ND	25.13	ND	>30	ND
Regorafenib	5.5	4.1–7.4	1.63	0.77–3.5	11.90	7.7–18.2	>30	ND	>30	ND

^1^ IC_50_ values determined by a three-parameter non-linear regression fit by non-linear least squares. ^2^ ND designates value not determined

## Data Availability

The datasets presented in this article are either included within the article and Appendix A or are not readily available due to technical and time limitations. Reasonable requests to access the datasets should be directed to the corresponding author, Dr. Christopher Coss.

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
