# Peer review of "Valine–Niclosamide for Treatment of Androgen Receptor Splice Variant-Positive Hepatocellular Carcinoma"

_cancers, 2025, doi:10.3390/cancers17152535_

Round 1
Reviewer 1 Report
Comments and Suggestions for Authors
In this study, niclosamide analogs were evaluated for their impact on AR protein in hepatocellular carcinoma cell lines with different androgen receptor phenotypes. Analogs were chosen based on improved bioavailability. These analogs exhibited anti-androgen receptor activity in in vitro and in vivo tests.
Methodology is appropriate and clearly described. The results support the conclusions.
A few minor editorial changes are needed -
Lines 38-39 – “Pharmacokinetic (PK) analyses were conducted to determine improvements in clearance and oral exposure. Amino-acid conjugates of niclosamide were developed to improve oral exposure and pharmacokinetic analyses were conducted to determine improvements.” – Statements seem redundant.
Line 130 – Should be Gainesville
Line 466 – Should be analogs
Line 478 – Should be Pharmacokinetics
Author Response
Comment 1: Lines 38-39 – “Pharmacokinetic (PK) analyses were conducted to determine improvements in clearance and oral exposure. Amino-acid conjugates of niclosamide were developed to improve oral exposure and pharmacokinetic analyses were conducted to determine improvements.” – Statements seem redundant.
Response 1: Lines 37-39 (Formerly 38-39) – Redundant sentences were combined. “Amino-acid conjugates of niclosamide were developed and pharmacokinetic (PK) analyses were conducted to determine improvements in clearance and oral exposure.”
Comment 2: Line 130 – Should be Gainesville
Response 2: Line 128 (Formerly 130) – Corrected from Gainsville to Gainesville
Comment 3: Line 466 – Should be analogs
Response 3: Line 475 (Formerly 466) – Corrected from anlaogs to analogs
Comment 4: Line 478 – Should be Pharmacokinetics
Response 4: Line 489 (Formerly 477) – Corrected from pharmacokinetics to Pharmacokinetics
Reviewer 2 Report
Comments and Suggestions for Authors
Valine-Niclosamide for Treatment of Androgen Receptor Splice Variant Positive
Hepatocellular Carcinoma
Overview
The study explored valine-niclosamide, a modified version of the anti-parasitic drug niclosamide, as a potential treatment for AR-SV-positive HCC. It demonstrated that valine conjugation improved niclosamide’s solubility and oral bioavailability, though further optimization is needed to enhance its conversion to the active form for effective in vivo efficacy.
Major Comments
1. Valine-niclosamide did not significantly reduce tumor cell viability in the HFA model, likely due to poor prodrug-to-parent conversion. Therefore, identifying this bottleneck highlights the need for optimizing cleavage mechanisms (e.g., alternative amino acid conjugates or enzymatic triggers), which could unlock its therapeutic potential.
2. Only 23 to 39% of valine-niclosamide converted to active niclosamide in vivo, limiting drug availability. This insight directs future work toward improving prodrug design (e.g., linker chemistry or esterase-targeted modifications) to enhance bioactivation. Please include a statement as limitation of the study or as future research scope.
3. The study did not explore valine-niclosamide with standard therapies (e.g., immune checkpoint inhibitors), despite its potential to enhance PD-1/PD-L1 blockade. Future combinatorial studies could position valine-niclosamide as a synergistic agent, leveraging its multi-pathway inhibition (AR, STAT3, NF-κB).
4. While GSEA implicated AR-SVs in oncogenic pathways (e.g., E2F, STAT3), direct
mechanistic links to niclosamide’s effects were not fully validated.
Minor Comments
1. Figure 1C is blurry.
2. The study broadly targeted AR-SVs but did not differentiate effects on specific variants, e.g., AR-V7, Line 107.
3. DMSO/Cremophor EL vehicle may influence drug solubility or tolerability in vivo. What about alternative formulations which could enhance bioavailability, please include brief discussion.
Remark
Overall, the work is well executed and the manuscript presentation is adequate. Minor adjustments should be made for further consideration.
Author Response
Major Comments
Comment 1: Valine-niclosamide did not significantly reduce tumor cell viability in the HFA model, likely due to poor prodrug-to-parent conversion. Therefore, identifying this bottleneck highlights the need for optimizing cleavage mechanisms (e.g., alternative amino acid conjugates or enzymatic triggers), which could unlock its therapeutic potential.
Response 1: Lines 783-787 – We added additional conclusions to address this bottleneck and highlight the need for future research into amino acid cleavage optimization.
Comment 2: Only 23 to 39% of valine-niclosamide converted to active niclosamide in vivo, limiting drug availability. This insight directs future work toward improving prodrug design (e.g., linker chemistry or esterase-targeted modifications) to enhance bioactivation. Please include a statement as limitation of the study or as future research scope.
Response 2: Lines 783-787 – We added additional conclusions to address opportunities to enhance bioactivation through alternative prodrug design.
Comment 3: The study did not explore valine-niclosamide with standard therapies (e.g., immune checkpoint inhibitors), despite its potential to enhance PD-1/PD-L1 blockade. Future combinatorial studies could position valine-niclosamide as a synergistic agent, leveraging its multi-pathway inhibition (AR, STAT3, NF-κB).
Response 3: Lines 796-798 – Clarified the need to evaluate valine-niclosamide and determine if it also is able to improve PD-1/PD-L1 blockage.
Comment 4: While GSEA implicated AR-SVs in oncogenic pathways (e.g., E2F, STAT3), direct mechanistic links to niclosamide’s effects were not fully validated.
Response 4: Lines 790-793 and 793-796 – Added to note that more mechanistic work is needed to validate if these are AR-SV mediated mechanisms and to note that Valine-Niclosamide impacts on these pathways need to be evaluated.
Minor Comments
Comment 1: Figure 1C is blurry.
Response 1: Line 401 – Figure 1C resolution limited to the resolution of the GSEA generated images but attempt to improve image quality was made and image was replaced.
Comment 2: The study broadly targeted AR-SVs but did not differentiate effects on specific variants, e.g., AR-V7, Line 107.
Response 2: Lines 775-780 – Added a caveat to note that the AR-SV makeup of HCC has not yet been fully determined and that a limitation of this study is that it broadly focuses on total AR-SVs rather than any specific variants.
Comment 3. DMSO/Cremophor EL vehicle may influence drug solubility or tolerability in vivo. What about alternative formulations which could enhance bioavailability, please include brief discussion.
Response 3: Selection of the DMSO/Cremophor EL-based vehicle used for all the in vivo experiments resulted from the evaluation of several published formulations culminating in identification of this vehicle as producing a clear solution at an appropriate concentration for IV niclosamide administration in the initial PK studies (Journal of Food and Drug Analysis, Vol. 14, No. 4, 2006, Pages 329-333). In other formulations tested, niclosamide was incompletely soluble, forming slurries or suspensions (5% Methocel A4M in water; 5% Tween 80/5% ethanol in saline (PMID: 27049719; PMID: 24740322)), or formed a solution containing a high proportion of organic solvent lacking translational compatibility (67% PEG400/33% N,N dimethylacetamide)(PMID: 26272032). While Cremophor-based drug formulations for oncology applications are still available, Cremophor-EL has been associated with toxicities and altered drug disposition (PMID: 11527683). Thus, continued pre-formulation efforts provide opportunities to potentially improve oral tolerance and bioavailability of niclosamide derivatives. This discussion was added to the Material and Methods Section, 2.10 Pharmacokinetic Studies and Parameter Analyses starting at line 258.